# OpenReview forum: "SAID: Empowering Large Language Models with Self-Activating Internal Defense"
_ICLR.cc/2026/Conference — ICLR 2026 Conference Withdrawn Submission_

### Official Review · Reviewer_FTV9 · 2025-10-16

**Soundness:** 3
**Presentation:** 3
**Contribution:** 2
**Rating:** 2
**Confidence:** 4

**Summary:**

This paper introduce Self-Activating Internal Defense (SAID), a training-free framework that repositions the defense process as a model-internal activation task. The method is evaluated across multiple LLMs and jailbreak attacks and shows strong defense performance.

**Strengths:**

1. The motivation and method are clear, and the visualization is helpful.

2. The authors provide sufficient evidence, including extensive experiments and ablation studies, to support the effectiveness of their proposed method.

3. This paper is well-written, making it easy to follow.

**Weaknesses:**

1. Since the primary effectiveness of jailbreak attacks stems from their ability to obfuscate an LLM’s intent perception, even after applying a simple distillation meta-prompt and segmentation, it is still highly likely that the attack can obscure the model’s understanding. This may cause the method to fail at the very first stage.

2. It appears that the authors generate a different candidate prefix pool for each target LLM. This process could be considered a form of information leakage. A fairer evaluation would be to generate the candidate pool using a white-box model, and then test the SAID framework on black-box commercial models, such as GPT and Gemini.

3. The time efficiency of SAID varies significantly across different target models, which seems unusual and warrants further explanation.

**Questions:**

Refer to weaknesses.

---

### Official Review · Reviewer_rdBB · 2025-10-25

**Soundness:** 1
**Presentation:** 1
**Contribution:** 1
**Rating:** 2
**Confidence:** 5

**Summary:**

The paper designs an LLM-defense called SAID based on prompt tuning. The evaluation is performed on 5 open-source LLMs against popular attacks such as GCG, AutoDAN, PAIR, etc.

**Strengths:**

* The paper tackles an important and challenging problem that is of interest to the ICLR audience.

**Weaknesses:**

* I had a hard time understanding the claims made in the paper, as it's not written for a technical audience. The presentation is inconsistent, incoherent, and non-rigorous, with the use of LLM-generated flowery language at various places (ICLR papers are relatively short; it is possible to create a strong submission without distracting LLM-generated text). Terms are used without being precisely defined. I dissect the writing issues one by one:
      1. Section 3.1 lists three defense objectives in plain English. Let's look at the first definition: "Robustness: Resist diverse and adaptive jailbreak strategies." What does resist mean? Will the model refuse or give out irrelevant output? What does the set of diverse and adaptive jailbreaks include? No defense will work against all possible attacks, so it's important to precisely characterize the class of attacks you are planning to handle. Similarly, the scope of "Utility Preservation" and "Time Efficiency" is undefined.

    2. Section 3.2.1 starts by mentioning "The initial stage of SAID tackles the challenge of intent obfuscation." What do you mean by intent obfuscation? Unless you precisely state what it means, one can claim anything as "intent obfuscation" and claim to address it.

    3. Line 196 mentions ", guiding the model to extract core semantic components", what counts as a core semantic component, and what does not? Next, the text says "This yields a distilled intent set S = {s1, s2, . . . , sk}, where each si ∈ Rti represents a concise intent." what counts as a concise intent and what does not? Are s1, s2, etc., single tokens, sentences, etc.?

   4. "For longer inputs that may obfuscate the model’s focus," what do you mean by model focus? The value of attention maps? At what layers? What do you mean by hierarchical distillation strategy?

   5. Can you mathematically characterize the Extract function used in (1)? Without rigorous specification, it is impossible to know if it is implemented as you require it to work.

  6. "We prepend an optimal safety prefix." What is the optimality criterion used here? Can you prove that your empirical search guarantees optimality?

  7. Eq. (3) and (4) are hard optimization problems. How do you solve them?

  8. "The final stage of SAID uses a conservative aggregation strategy based on risk minimization. " What counts as conservative aggregation and what does not? Where is the notion of risk minimization defined?

* The approach has limited novelty. The paper overclaims that it is introducing a new paradigm for LLM defense. In reality, the method appears to be doing a form of prompt tuning.

* I believe that the authors should report the performance of their method on priming/prefilling attacks (https://arxiv.org/abs/2312.12321, https://arxiv.org/abs/2404.02151) as done by Deep Safety Alignment (https://arxiv.org/pdf/2406.05946).

* The experimental evaluation does not provide any evidence or mechanistic insights that any type of intent distillation happens or that there is any change in model focus, as claimed in the paper.

**Questions:**

To make this paper ready for publication at a major conference, a serious effort is needed to improve the quality of writing. I don't have any questions for the authors that would sway my opinion, but I encourage them to consider incorporating my suggestions to improve the paper.

---

### Official Review · Reviewer_dC9y · 2025-11-03

**Soundness:** 2
**Presentation:** 2
**Contribution:** 2
**Rating:** 4
**Confidence:** 4

**Summary:**

This work proposes SAID, a framework that reframes jailbreak defense as internal capability activation via three stages: model-native intent distillation, optimal safety-prefix probing, and conservative aggregation. Across some open-sourced LLMs and representative jailbreaks, SAID moderately reduces harmfulness while preserving benign utility.

**Strengths:**

- Trendy Topic
- Well-Structured Manuscript

**Weaknesses:**

- Insufficient Experimental Coverage
- Unclear Description

**Questions:**

- In the final stage, the aggregation strategy used is conservative. Are there other strategies that can be adopted? Is the current conservative aggregation optimal?
- For the evaluated attacks/defenses, the manuscript only briefly describes their workflows without providing specific settings. Providing detailed settings is crucial for fair evaluation and comparison. Besides, can SAID resist adaptive attacks, such as splitting a harmful problem into multiple harmless subproblems?
- Recently, various guardrails (such as OpenAI Moderation and LLaMA-Guard-3/4) have been widely used. I am curious about whether SAID has an advantage in defensive performance compared to them.
- From which dataset do the user inputs used to find the optimal prefix come? Do they overlap with the samples used in the evaluation?
- How does SAID behave across long conversations, tool calls, or retrieval-augmented contexts where harmful intent emerges gradually?
- What are the per-query overheads for distillation and probing under typical context lengths, especially compared with other baselines?
- In Section 4.2.2, the description is too vague to demonstrate whether (and to what extent) the proposed SAID has an advantage in utility and efficiency compared to other methods.
- In Table 3, why are only four attacks evaluated instead of six for LLaMA3-8B?
- In Tables 2/4, why are the evaluations on LLaMA3-8B (and Vicuna-13B) omitted?
- Minors:
  - What does "prefix-based causal profiling" mean in the Introduction?
  - In Section 3.1, "intents detection."
  - In Section 3.2, "Fig. 2."
  - In Section 4.1.3, "Just-Eval (Lin et al., 2023), 000."

---

### Official Review · Reviewer_NJAR · 2025-11-03

**Soundness:** 2
**Presentation:** 2
**Contribution:** 2
**Rating:** 2
**Confidence:** 4

**Summary:**

This paper proposes a prompting based defense for jailbreaks where the same model is first used to extract intents from the user query, then classify them into benign or malicious, and finally refuse to respond if any of the classifications were malicious.

**Strengths:**

N/A

**Weaknesses:**

- The accuracy of the intent extraction is unclear and not evaluated.
- The paper doesn't clearly mention how the optimal prefix tuning is done. What dataset was used to find/tune the optimal prefixes for the defense and what dataset was used to evaluate the defense? -- these should ideally be different sources/distributions to test for generalization.
- The defense hasn't been evaluated against adaptive attacks. An attack that is aware of the intent extraction/classification might be able to target those components and bypass the defense. For instance, since the extraction/classification is done using an LLM, they are prone to prompt injection attacks.

**Questions:**

N/A

---

### Official Review · Reviewer_ehxQ · 2025-11-05

**Soundness:** 3
**Presentation:** 3
**Contribution:** 3
**Rating:** 6
**Confidence:** 3

**Summary:**

The paper aims to mitigate LLM jailbreaking threats, and introduces Self-Activating Internal Defense (SAID), a training-free paradigm that reframes defense as an internal capability activation rather than external correction. SAID utilizes the LLM's reasoning abilities through a three-stage pipeline: model-native intent distillation to extract core semantics, optimal safety prefix probing to trigger latent safety awareness, and a conservative aggregation strategy for final decision-making. Experimental results on five open-source LLMs against six jailbreak attacks indicate that SAID outperforms state-of-the-art defenses in reducing harmful outputs while maintaining performance on benign tasks with minimal reported computational overhead.

**Strengths:**

* Novel perspective: I like their distinct approach of shifting from external filters or resource-intensive fine-tuning to activating the model's intrinsic safety mechanisms without parameter updates.
* Thorough, extensive experiments: their evaluations show that SAID consistently achieves lower average harmful scores compared to varied baseline defenses across various LLMs and jailbreak attacks. The paper also includes ablation studies that validate the individual contributions of the intent distillation, safety probing, and aggregation stages to the overall defense effectiveness. Additionally, the analysis of prefix variants in sec 4.2.3 provides evidence that "semantically neutral" prefixes are more effective for balancing safety and utility than instructional ones.

**Weaknesses:**

* Lack of clarity and scalability in optimal prefix selection: While the concept of an optimal safety prefix is central to the approach, the practical process used to identify it seems to be under-specified. The methodology describes a structured empirical search starting with handcrafted examples and uses few-shot prompting to generate a candidate pool (Sec. 3.2.2). Equ4 frames this as an optimization problem but does not detail the search algorithm used to solve it. This reliance on manual crafting and undefined search procedures hinders reproducibility and I feel the work could be strengthened by providing a fully reproducible algorithmic description of the current search process or exploring automated alternatives.

* Reliance on brittle heuristics for final decision: the final Conservative Aggregation stage relies on a binary `IsRefusal( )` function to determine if the model has rejected a probed intent (Eq5). Appendix C details that this is implemented using a static ''Harmful Words List" containing string matches like ''I cannot", ''I'm sorry", etc.
But I guess there's a chance for such heuristic approach to get brittle; \eg sophisticated attacks could manipulate the model to generate harmful content while avoiding these specific standard refusal triggers, etc.

**Questions:**

* Reproducibility of prefix optimization: Can you provide precise details on the structured empirical search used to solve Equ4 for finding the optimal prefix?

* Robustness of refusal detection: Regarding the final refusal detection mechanism, how robust is the reliance on the exact string matching of the ``Harmful Words List''? Did you evaluate scenarios where an attack successfully prompts the model to generate harmful content without using these standard refusal phrases (i.e., bypassing the final check), or scenarios where benign outputs accidentally trigger these phrases?

---

### Note · Authors · 2025-11-12

**Comment:**

We have carefully considered the insightful comments provided by the reviewers and acknowledge that our work requires additional refinement and further experiments. We are grateful for the reviewers’ time, effort, and constructive feedback, which will help us improve our research.
Accordingly, we would like to withdraw our submission at this time.

**Withdrawal Confirmation:**

I have read and agree with the venue's withdrawal policy on behalf of myself and my co-authors.